# The Variability of Tryptophan Metabolism in Patients with Mixed Type of Irritable Bowel Syndrome

**DOI:** 10.3390/ijms25052550

**Published:** 2024-02-22

**Authors:** Jan Chojnacki, Paulina Konrad, Marta Mędrek-Socha, Aleksandra Kaczka, Aleksandra Błońska, Radosław Zajdel, Cezary Chojnacki, Anita Gąsiorowska

**Affiliations:** 1Department of Clinical Nutrition and Gastroenterological Diagnostics, Medical University of Lodz, 90-647 Lodz, Poland; paulina.konrad@umed.lodz.pl (P.K.); marta.medrek-socha@umed.lodz.pl (M.M.-S.); aleksandra.kaczka@umed.lodz.pl (A.K.); aleksandra.blonska@umed.lodz.pl (A.B.); cezary.chojnacki@umed.lodz.pl (C.C.); 2Department of Computer Science in Economics, University of Lodz, 90-255 Lodz, Poland; radoslaw.zajdel@uni.lodz.pl; 3Department of Gastroenterology, Medical University of Lodz, 92-213 Lodz, Poland; anita.gasiorowska@umed.lodz.pl

**Keywords:** irritable bowel syndrome, tryptophan metabolism, gut microbiome

## Abstract

Patients with a mixed type of irritable bowel syndrome (IBS-M) experience constipation and diarrhea, which alternate between weeks or months. The pathogenesis of this syndrome is still little understood. The aim of the study was mainly to evaluate the urinary excretion of selected tryptophan (TRP) metabolites during the constipation and diarrhea periods of this syndrome. In 36 patients with IBS-M and 36 healthy people, serum serotonin level was measured by ELISA and urinary levels of 5-hydroxyindoleacetic acid (5-HIAA), kynurenine (KYN) and indican (3-IS) were determined using the LC-MS/MS method. The levels of all above metabolites were higher in the patient group, and increased significantly during the diarrheal period of IBS-M. In particular, the changes concerned 5-HIAA (3.67 ± 0.86 vs. 4.59 ± 0.95 mg/gCr, *p* < 0.001) and 3-IS (80.2 ± 17.4 vs. 93.7 ± 25.1 mg/g/Cr, *p* < 0.001). These changes coexisted with gut microbiome changes, assessed using hydrogen-methane and ammonia breath tests. In conclusion, the variability of TRP metabolism and the gut microbiome may cause the alternation of IBS-M symptoms.

## 1. Introduction

Irritable bowel syndrome (IBS) is a common functional disorder of the gastrointestinal tract with a varied clinical picture. According to the Rome IV Criteria, four types of IBS are distinguished: that with predominant constipation (IBS-C), predominant diarrhea (IBS-D), mixed bowel habits (IBS-M) and unclassified type (IBS-U) [1]. The clinical boundaries between them are not precisely defined and not permanent. It is considered that about one half of patients with IBS can change subtype over the years [2]. However, some patients experience constipation and diarrhea, which alternate between weeks or months [3]. These patients may qualify as having IBS-M. The Rome Criteria requires that symptoms be present for at least three months, with onset at least six months prior to diagnosis. People who have IBS-M have all the symptoms associated with other types of IBS, including abdominal pain, a filling of incomplete evacuation, changes in bowel frequency and in stool consistency, mucus in the stool, gas and bloating [4]. The pathogenesis of IBS-M is multifactorial and still little understood. These factors include gut dysmotility, visceral hypersensitivity, dysfunction of the gut–brain axis, dysbiosis, etc. [5]. It is estimated that the majority of IBS patients experience changes in the gut microbiome [6]. Although it is a functional disease, the involvement of subclinical inflammatory factors cannot be ruled out. Low-grade immune activation may exist in IBS-M. Bacterial and food endotoxin could play a role in the immunological response and may damage the intestinal barrier [7,8]. Intestinal bacteria can exert an influence directly or by participating in the metabolism of dietary ingredients [9,10]. The first line of treatment for IBS is an appropriate diet [11,12,13,14]. Numerous patients with IBS attribute their symptoms to food, which can cause exaggerated complaints. Currently, the low FODMAP diet is often used to treat all types of IBS [15,16,17]. Every diet contains various nutrients, which may have beneficial or unsatisfactory effects. The amounts of consumed ingredients and their metabolism is also important. An example of this is the intake of tryptophan (TRP), which is over 90 percent metabolized in the intestines, on the serotonin, kynurenine and indole pathways. Among them, serotonin plays a crucial role in the regulation of gastrointestinal functions [18,19]. It can result in smooth muscle concentration or relaxation through various receptors. The level of serotonin depends on the amount of tryptophan consumed, as well as on the activity of its metabolic pathways. The serotonin pathway is initiated by endogenous tryptophan hydroxylase (TPH-1), and the kynurenine pathway by indoleamine 2,3-dioxygenase (IDO-1). Bacterial enzymes called tryptophanase (TNA) initiate tryptophan metabolism via the indole pathway. These enzymes compete for access to ingested TRP [20]. The balance of these pathways can be altered due to many factors, including dysbiosis [21,22]. It is not known whether fluctuation in tryptophan metabolism may influence the clinical picture of IBS-M.

The aim of the study was to evaluate the urinary excretion of selected TRP metabolites during the constipation and diarrhea periods of the mixed type of IBS, focusing on the serotonin pathway.

## 2. Results

Participants were correctly selected in terms of age, gender and nutritional status. The results of routine laboratory tests were also similar. Both groups differ significantly in the levels of C-reactive protein and fecal calprotectin, which may indicate low-grade inflammation of the intestinal mucosa (Table 1).

Basal hydrogen concentration in exhaled air was higher in the diarrhea period of IBS-M. Similar differences were found at 90 min. Differences between these time intervals above 20 ppm were found in 6 patients in group IIa and in 14 patients in group IIb.

Methane concentration was higher in group IIb only in the 150th minute of the test.

Ammonia concentration was higher in group II b compared to group IIa in all time intervals (Table 2).

Serotonin levels in the serum during diarrhea were higher compared to constipation (*p* = 0.003), but the urinary excretion of TRP was not significantly different (Table 3, Figure 1).

Urinary levels of 5-HIAA during diarrhea were also significantly higher than during constipation (*p* < 0.001, Table 3, Figure 1).

The differences in results of urinary levels of kynurenine obtained during constipation and diarrheal periods were also significant (*p* = 0.005, Table 3, Figure 2).

Urinary concentration of indican in the diarrhea period was significantly higher than in the constipation period (*p* < 0.001, Table 3).

The levels of all above TRP metabolites in both groups of IBS-M patients were significantly higher than in the group of healthy people (Table 3).

## 3. Discussion

The obtained results indicate the variability of TRP metabolism in patients with IBS-M. The changes affect all of its metabolic pathways, but especially serotonin secretion. During the constipation phase, the serum serotonin level was slightly higher than in healthy people, and significantly increased during the period of diarrhea. A similar direction of changes applies to urinary 5-hydroxyindoleacetic acid (5-HIAA) excretion. These changes cannot be compared to the results obtained by other researchers because the studies were conducted mainly in patients with IBC-D or IBS-C. Most often, it has been reported that an increase in serotonin secretion occurs in the diarrheal type of this syndrome [23,24,25]. These changes are associated with an increase in the number enterochromaffin cells in the colonic mucosa [26,27]. In contrast, other researchers reported that serum increase of 5-HT was related to constipation [28,29,30]. The discrepancy in results is probably caused by the complex effects of serotonin on the motor activity of the GI tract. 5-HT can activate cholinergic excitatory neurons mediated by 5-HT_3_ and 5-HT_4_ receptors or nitric oxide inhibitory enteric motor neurons mediated by 5-HT_4_, 5-HT_1a_ or 5-HT_1D._ However, receptor 5-HT_4_ stimulation led to the relaxation of the distal colon [31,32]. These differences may also be due to receptor topography and balance between its activation and desensitization [33]. All of the above factors may influence the clinical assessment of serotonin effects. Disorders of serotonin synthesis are important in inflammatory and functional diseases. Pro-inflammatory factors, including bacteria and viruses, cause its increased secretion [34,35]. In these processes, changes in the gut microbiome are important. They may concern the number of bacteria or their composition, but research results are mixed [36]. In general, it is believed that changes in IBS patients showed a significant increase in Enterobacteriaceae (Proteobacteria phylum) and in Bacteroides (Bacteroidetes phylum), while Faecalibacterium (fermicutes phylum) and Bifidobacterium (Actinobacteria phylum) are decreased compared to healthy people [37,38]. The profile of bacteria is very complex, and their properties are still not significantly known. Quantitative changes are also difficult to assess. For this reason, a non-invasive assessment of bacterial metabolites is more often used in research.

Our study showed that the results of the hydrogen-methane test change over time. In the period of constipation, only some patients showed small intestinal bowel overgrowth. However, during the diarrheal period, these criteria were present in more than half of the patients. The concentration of methane in the exhaled air was similar in both phases of IBS, which does not exclude the participation of this component in the regulation of intestinal motility in some patients [3].

Changes in the number of bacteria in the intestines were accompanied by disturbances in tryptophan metabolism, mainly the serotonin pathway. Microbiota regulates 5-HT production via several mechanisms, by increasing TPH-1 activity or by regulating SERT activity [22,39,40]. Bacteria and pro-inflammatory cytokines also activate the kynurenine pathway, but the role of its metabolites in the pathogenesis of IBS is little known. Higher levels of plasma kynurenine were found in these patients [41,42,43,44], but these changes were related to accompanying mental disorders.

Both serotonin and kynurenine also have a considerable influence on gut microbiota, and alternations in their homeostasis are implicated in the pathogenesis of many gastrointestinal disorders [45].

Indican is also an important metabolite of TRP. It is synthesized via tryptophanase—expressing colon anaerobic bacteria on the pathway of L-tryptophan—indole-indoxyl—3-indole sulfate. This metabolite is considered a quantitative biomarker of the state of intestinal microbiome [46,47]. Indican exerts anti-inflammatory activities, but its toxic influences cannot be excluded [48,49]. In our material, the urinary excretion of indican increased during diarrhea, as did the concentration of hydrogen ions in the exhaled air.

The indirect and direct involvement of gut bacteria in tryptophan metabolism has been well documented in numerous previous and current studies [50,51,52]. Similarly, the role of bacteria in the pathogenesis of IBS is recognized. For example, bacterial overgrowth of the small intestine (SIBO) is diagnosed in about half patients with this syndrome. On the other hand, the clinical picture of SIBO varies individually, with chronic diarrhea in some patients and constipation in others. In addition, about 10% of patients with a diagnosis of SOBO show none of gastrointestinal symptoms. The causes of recurrence and variability of symptoms are also unclear and need further research, especially regarding IBS-M, as there are very few publications on this topic.

An increase in the number of bacteria does not rule out changes in the microbiome profile. The reasons for these changes are not clear. Dietary factors have a significant impact on the microbiome [6,53,54]. Our patients had the same diet with limited TRP intake, which requires that other factors are noted [55]. It can be assumed that people with IBS-M constantly experience changes in their gut microbiome. It may be that, during the diarrheal phase, there is an increased evacuation of bacteria from the digestive tract. When accepting such a thesis, the treatment should take into account the factors contributing to the re-growth of intestinal bacteria. For this purpose, cyclic use of selected antibiotics is recommended, but their effect may cause secondary dysbiosis. The changing nature of bowel symptoms makes it difficult to choose a good treatment strategy. Some probiotics have beneficial effects as they change the microbiome profile [56,57,58]. Dietary nutrients, consumed in optimal amounts, can have a similar effect [36,59]. In particular, functional diarrhea or constipation can be reduced by limiting or increasing the intake of tryptophan [60,61].

Our study has some limitations. Firstly, the severity of FC symptoms is based mainly on the patients’ feelings. These criteria apply to all functional diseases of the gastrointestinal tract. Secondly, the assessment of nutrition quality also depends on the patients’ complaints, but good cooperation with dietitians ensures the reliability of the results. Moreover, the state of the microbiome was determined using breathing tests, but the direction of changes in the concentration of all those tested was similar, which increases their value.

In conclusion, our results confirm that, in IBS-M patients, there are changes in the gut microbiome, which secondarily cause variability in tryptophan metabolism. These changes should be taken into account in dietary treatment, recommending optimal tryptophan intake. The key to achieving good results of treatment is to better understand the intestinal microbiome and properties of individual bacteria, including their impact on the metabolism of nutrients.

## 4. Material and Methods

### 4.1. Participants

The study included 40 subjects without any abdominal complaints (Group I, Controls) and 40 patients with the mixed type of irritable bowel syndrome (Group II, IBS-M), aged 31–64 years. In total, 36 people in each group completed the research program. The duration of abdominal symptoms ranged from 3 to 11 years. The research was conducted between 2019 and 2023. Patients were diagnosed according to the Rome IV Criteria [1], taking into account the following symptoms: abdominal pain, changes in bowel frequency, hard or loose stools, bloating, passing gas, visible distension and presence of mucus in the stool. The severity of abdominal symptoms was assessed and scored on a scale of 1 to 7 points, based on what the patients wrote in their diary every day for three months during both the constipation and diarrhea period. Previously, patients were provided with appropriate instructions and dietary recommendations by healthcare professionals. Patients could only use laxatives or loperamide. Diagnostic tests were performed in the fifth week of the constipation period and then in the fifth week of the diarrhea period. Tests for individual patients were repeated in the same calendar year.

### 4.2. Diagnostic Procedures

To exclude inflammatory and cancerous diseases of the large intestine, all patients underwent endoscopic and histological examination of colonic mucosa.

Inclusion criteria: the study included patients who met the Rome IV Criteria and had severe constipation or diarrhea above six points and intensity of all abdominal symptoms above 21 points.

Exclusion criteria: inflammatory bowel disease, allergy and food intolerance, liver and renal diseases, diabetes, severe anxiety or depression, and the use antibiotics, probiotics, or psychotropic and antispasmodics drugs in the month prior to enrolment in the study.

### 4.3. Breathing Test

The hydrogen-methane breath test (HBT) was performed using Gastrolyzer (Bedfont, Ltd., Harrietsham, UK). At the beginning, the fasting levels of hydrogen and methane in breath were measured. Then, 10 g of lactulose dissolved in 200 mL of water was administrated to the patients, and breath samples were collected immediately and at 15 min intervals for 3 h. The criterion for a positive SIBO diagnosis was a minimum increase of 20 ppm of hydrogen within the first 90 min of testing.

The ammonia breath test (ABH) was performed using a gas analyzer (HELIC ABT Reader, AMA Co Ltd, Mikkeli, Finlkand), after discontinuing antibiotics and probiotics for 4 weeks and medications inhibiting gastric secretion, alkaline drugs and alcohol for 2 weeks. On the day of the examination, the concentration of ammonia (NH3) in the expiratory air was determined on an empty stomach and then at 15 min intervals for 3 h after ingestion of 250 mL of protein solution (Nutridrink Protein—Nutricia).

Both tests, i.e., HBT and ABT, were performed in the same week, after discontinuing antibiotics and probiotics for 2 weeks. The concentration of the above ions (hydrogen, methane, ammonia) at 0, 90 and 150 min of the tests was assumed for statistical analysis.

### 4.4. Laboratory Tests

The following routine laboratory tests were performed in all subjects: blood cell count, quantification of protein, glucose, glycated hemoglobin, profile of lipids, bilirubin, iron, urea, creatinine, thyroid stimulating hormone, free thyroxine, free triiodothyronine antibodies to tissue transglutaminase, the activity of alanine and asparagine aminotransferase, alkaline phosphatase, gamma-glutamyltranspeptidase, amylase and lipase and deaminated gliadin peptide. The serum concentration of C-reactive protein (CRP) was determined using a latex agglutination photometric assay in COBAS INTEGRA 800 (Roche Diagnostic, Basel, Switzerland). Fecal calprotectin (FC) was evaluated using a sandwich ELISA test in a Quantum Blue Reader (Buhlmann Diagnostics, Amherst, NH, USA).

The serum serotonin concentration was measured with the ELISA method using an Immuno Biochemical Laboratories kit (No. RE 59121). The measurements were performed using photometry at a wavelength of 450 nm (Expert 96—Reader-Biogenet, Warsaw, Poland).

Urine samples for TRP and its metabolite testing were collected in the morning on an empty stomach into a special container with a 0.1% hydrochloric acid solution as a stabilizer. We determined the concentration of TRP and its following metabolites: 5-hydroxyindoleacetic acid (5-HIAA), kynurenine (KYN) and 3-indoxyl sulfate (Indican) using liquid chromatography with tandem mass spectrometry (LC–MS/MS in accordance with the manufacturer instructions (Ganzimmun Diagnostics AG, Mainz, Germany; D-ML-13147-01-01, accepted by the European Parliament—No 765/2008). The levels of these metabolites were expressed in mg per gram of creatinine (mg/gCr). The above metabolites of tryptophan were considered as exponents of the activity of serotonin (5-HIAA), kynurenine (KYN) and indole (Indican) pathways. All laboratory materials were collected on the same day.

### 4.5. Nutritional Intervention

It was recommended that all individuals maintain their current diet, but with limited tryptophan intake at 1200 mg daily. Patients recorded the type and quantity of products consumed every day for 30 days in a nutritional diary for investigation purposes. The average daily TRP intake was calculated using the application Kcalmar.pro-Premium (Hermex, Lublin, Poland). However, the patients applied a balanced diet with a total caloric value of 2000 kcal and with a minimum daily intake of 50 g of protein, 270 g of carbohydrates and 70 g of fats. The day before the examination, everyone was provided with the same diet with the TRP content calculated earlier. The content of TRP in food products was determined according to the findings of the Polish National Institute of Public Health. Optimal amounts of protein, carbohydrates and fats, as well as fiber, were maintained. The use of any drugs was forbidden, except for laxatives and loperamide. It was recommended that patients complete a daily diet diary, under the control of nutritionists, with whom they had telephone and e-mail contact. Dietary instruction was also provided to patients’ families and caregivers. After each week, the amount of TRP intake was analyzed to evaluate compliance with the recommendations. Follow-up medical examinations with the assessment of the somatic and mood symptoms were performed every two weeks.

The research was conducted using an open-label clinical trial. Written consent was obtained from all participants. The study was conducted according to the guidelines of the Declaration of Helsinki and the Guidelines for Good Clinical Practice.

### 4.6. Data Analysis

A test of the normality of the distribution of the variables under study and the homogeneity of the variance was performed. The Shapiro–Wilk test and Levene’s test showed that the distributions of the data differed (*p* < 0.05 and *p* > 0.05 for some subgroups), thus making it impossible to use parametric tests in the statistical analysis. Thus, the Wilcoxon ANOVA test, the equivalent of Student’s *t*-test for related data, was used.

## 5. Conclusions

The variability of tryptophan metabolism and the gut microbiome coexist in patients with IBS-M, and may cause symptom alternation, which should be taken into account in the treatment strategy for this syndrome.

## Figures and Tables

**Figure 1 ijms-25-02550-f001:**
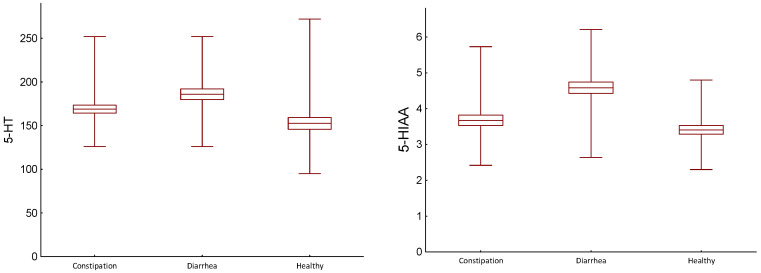
Comparison of serum serotonin (5-HT) levels (ng/mL) and urinary levels of 5-hydroxyindoleacetic acid (5-HIAA, mg/g/Cr) in healthy people and IBS-M patients during constipation and diarrhea; differences between controls and patients were evaluated using a Wilcoxon test; center line—mean value; box—SD; whiskers—min-max.

**Figure 2 ijms-25-02550-f002:**
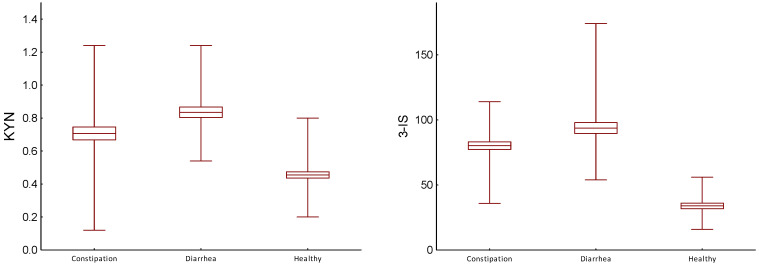
Comparison of urinary levels of kynurenine (KYN, mg/gCr) and indican (3-IS, mg/gCr) in healthy people and IBS-M patients during constipation and diarrhea period; differences between controls and patients were evaluated by Wilcoxon test; center line—mean value; box—SD; whiskers—min-max.

**Table 1 ijms-25-02550-t001:** General characteristics and the selected biochemical blood parameters in healthy subjects (Group I, Controls) and in patients with the mixed type of irritable bowel syndrome (IBS-M) in the constipation period (Group IIa) and in the diarrhea period (Group IIb).

Feature	Group I (*n* = 36)	Group IIa (*n* = 36)	Group IIb (*n* = 36)	*p*
Age (years)	45.4 ± 9.4.	44.7 ± 11.3	44.7 ± 11.3	ns
Gender M/F	8/28	7/29	7/29	ns
BMI (kg/m^2^)	23.8 ± 1.3	24.1 ± 2.2	24.0 ± 2.3	ns
GFR (mL/min)	99.8 ± 4.4	97.2 ± 6.8	98.5 ± 4.8	ns
ALT (µ/L)	13.5 ± 2.8	16.8 ± 3.7	16.2 ± 4.2	ns
AST (µ/L)	11.9 ± 1.8	12.1 ± 2.8	12.9 ± 4.1	ns
CRP (mg/L)	2.8 ± 1.9	3.4 ± 2.7	6.3 ± 3.2	<0.01 ^bc^
FC (µg/g)	11.8 ± 7.5	25.6 ± 12.8	42.8 ± 15.8	<0.01 ^abc^

GFR—glomerular filtration rate; ALT—alanine aminotransferase; AST—aspartate aminotransferase; CRP—C-reactive protein; FC—fecal calprotectin; data are presented as average ± SD; differences between groups were assessed using Student’s *t*-test; ^a^—I vs. IIa; ^b^—I vs. IIb; ^c^—IIa vs. IIb.; ns-not statistically significant.

**Table 2 ijms-25-02550-t002:** Concentrations of hydrogen, methane and ammonia in exhaled air in IBS-M patients in constipation period (Group IIa) and in diarrhea period (Group IIb) determined at 0, 90 and 150 min with the hydrogen/methane test and ammonia breath test.

Ions (Time, min)	Group IIa (ppm)	Group IIb (ppm)	*p*
Hydrogen (0)	6.46 ± 3.54	12.9 ± 11.86	<0.05
Hydrogen (90)	23.1 ± 5.72	37.5 ± 11.2	<0.01
Hydrogen (150)	91.4 ± 18.3	98.4 ± 26.7	ns
Methane (0)	4.5 ± 1.6	4.8 ± 1.4	ns
Methane (90)	4.6 ± 1.8	5.2 ± 1.6	ns
Methane (150)	11.1 ± 4.2	15.2 ± 5.7	<0.05
Ammonia (0)	5.2 ± 3.1	8.3 ± 2.8	<0.001
Ammonia (90)	6.1 ± 2.9	9.4 ± 3.9	<0.01
Ammonia (150)	10.3 ± 3.1	16.6 ± 4.1	<0.001

Differences between subgroups were assessed using Student’s *t*-test; ns—non-significant (*p* > 0.05).

**Table 3 ijms-25-02550-t003:** Comparison of serum serotonin (5-HT) levels (ng/mL) and urinary levels of tryptophan (TRP), 5-hydroxyindoleacetic acid (5-HIAA), kynurenine (KYN) and indican (3-IS) in healthy people and IBS-M patients during constipation and diarrhea periods.

Parameter	Group (H) (D) (C)	Mean/SD/ Median/IQR	Statistical Analysis, *p*, Wilcoxon Test	
5-HT (ng/mL)	Constipation (C)	168.86/27.404 161.50/29.00	D/C	0.003
	Diarrhea (D)	185.83/36.773 170.00/65.00	H/D	0.002
	Healthy (H)	152.56/40.841 147.00/57.00	H/C	0.045
TRP (mg/gCr)	Constipation (C)	11.62/1.654 11.80/2.250	D/C	0.273
	Diarrhea (D)	11.64/1.625 11.80/2.25	H/D	0.001
	Healthy (H)	13.82/1.857 14.20/2.65	H/C	0.001
5-HIAA (mg/gCr)	Constipation (C)	3.67/0.862 3.64/1.035	D/C	0.001
	Diarrhea (D)	4.59/0.953 4.47/1.51	H/D	0.001
	Healthy (H)	3.41/0.735 3.20/0.80	H/C	0.081
KYN (mg/gCr)	Constipation (C)	0.71/0.235 0.72/0.255	H/C	0.005
	Diarrhea (D)	0.84/0.189 0.82/0.25	H/D	0.001
	Healthy (H)	0.46/0.113 0.50/0.10	H/C	0.001
3-IS (mg/gCr)	Constipation (C)	80.28/17.491 81.50/17.50	D/C	0.001
	Diarrhea (D)	93.75/25.166 86.00/29.50	H/D	0.001
	Healthy (H)	33.97/12.557 35.00/22.50	H/C	0.001

## Data Availability

The data supporting the reported results can be provided by the corresponding author on reasonable request.

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
