# Peer review of "The Variability of Tryptophan Metabolism in Patients with Mixed Type of Irritable Bowel Syndrome"

_ijms, 2024, doi:10.3390/ijms25052550_

Round 1

Reviewer 1 Report

Comments and Suggestions for Authors

The authors of this study focused on the evaluation of urinary excretion of selected tryptophan 17 (TRP) metabolites during constipation and diarrhea period, in patients with irritable bowel syndrome.

1.    The results of this study are not relevant, as the number of patients enrolled is too small,  the criteria of diagnosis are not sufficient, and statistical analysis is not significant.

2.    Most of the neuroendocrine tumors (NET) produce and secrete a large number of peptide hormones and amines, causing specific clinical syndromes, such as  carcinoid syndrome,  diarrhhea being a common symptom. Basal and/or stimulated levels of urinary-5-HIAA represent a specific marker for these syndromes. NET could be a challenging differential diagnosis that was not excluded in this study.

3.    Discussions must be improved, adding references according to recent studies. It is suggested that there is a relationship between IBS and impaired tryptophan metabolism  /10.1016/j.chom.2018.05.003.   Kynurenine levels and the ratio of kynurenine to tryptophan in plasma were shown to be increased in IBS patients compared to healthy patients, indicating increased activity of the kynurenine pathway in IBS patients. However, in line with the importance of serotonin for the regulation of GI motility, a study including both constipation-predominant and diarrhea-predominant IBS patients reports dysfunctional serotonin synthesis and transport, with decreased expressions of TPH1 and SERT in the colorectal biopsies of IBS patients10.1152/ajpgi.00153.2011. Serotonin levels differ between IBS subtypes,  being decreased in constipation-predominant IBS, and increased in diarrhea-predominant IBS, according to a previous study 10.1038/.2012.8

 Since the serotonin system can be directly modulated by the gut microbiota  10.1073/pnas.1720017115; 10.1016/j.cell.2015.02.047 further research is needed on whether dysregulation of the serotonin system in IBS patients may be partly due to alterations of the gut microbiota.

Comments on the Quality of English Language

Moderate English editing is required

Author Response

Comment:    The results of this study are not relevant, as the number of patients enrolled is too small,  the criteria of diagnosis are not sufficient, and statistical analysis is not significant.

Response : The results obtained have important clinical implications. The Rome IV Criteria are universally recognized. Mixed type of IBS is not as common as other (IBS-D, IBS-C), and it cause great diagnostic and therapeutic difficulties. The results were assessed by an experienced statistician  and our co-author (see the list – RZ).

Comment:  Most of the neuroendocrine tumors (NET) produce and secrete a large number of peptide hormones and amines, causing specific clinical syndromes, such as  carcinoid syndrome,  diarrhoea being a common symptom. Basal and/or stimulated levels of urinary-5-HIAA represent a specific marker for these syndromes. NET could be a challenging differential diagnosis that was not excluded in this study.

Response: In several years of examination of our patients, NET and many other diseases causing diarrhoea or constipation were excluded.

Comment:  Discussions must be improved, adding references according to recent studies. It is suggested that there is a relationship between IBS and impaired tryptophan metabolism  /10.1016/j.chom.2018.05.003.   Kynurenine levels and the ratio of kynurenine to tryptophan in plasma were shown to be increased in IBS patients compared to healthy patients, indicating increased activity of the kynurenine pathway in IBS patients. However, in line with the importance of serotonin for the regulation of GI motility, a study including both constipation-predominant and diarrhoea-predominant IBS patients reports dysfunctional serotonin synthesis and transport, with decreased expressions of TPH1 and SERT in the colorectal biopsies of IBS patients10.1152/ajpgi.00153.2011. Serotonin levels differ between IBS subtypes,  being decreased in constipation-predominant IBS, and increased in diarrhoea-predominant IBS, according to a previous study 10.1038/.2012.8

Response: The direct and indirect involvement of gut bacteria in tryptophan metabolism has been well documented in numerous  previous and current studies ( some of the indicated articles  were  included in the list of references -50,51,52).  Similarly, the role of microbiome in the pathogenesis of IBS is recognized. For example, bacterial overgrowth of the small intestine (SIBO) is diagnosed about in half of patients   with IBS. On the other hand,  the clinical picture of SIBO varies individually, with chronic diarrhoea in some patients, and constipation in other. In addition, about 10% of patients with SIBO do show none of gastrointestinal symptoms. The cause of recurrence and variability of symptoms are still unclear.  Further research is needed especially regarding IBS-M, as there are very few publications on this topic.

Reviewer 2 Report

Comments and Suggestions for Authors

As a reviewer, I find this study on the urinary excretion of selected tryptophan (TRP) metabolites during constipation and diarrhoea periods in patients with mixed-type irritable bowel syndrome (IBS-M) to be both timely and informative. The investigation sheds light on the potential role of TRP metabolism and gut microbiome alterations in the pathogenesis of IBS-M symptoms. The use of robust methodologies, such as ELISA and LC-MS/MS, adds credibility to the findings. However, further clarification on the specific mechanisms linking TRP metabolism, gut microbiome changes, and symptom alternation in IBS-M would greatly enhance the impact of the study. Additionally, providing insights into potential therapeutic implications stemming from these findings could make the study more clinically relevant. Overall, the study offers valuable insights into the complex interplay between TRP metabolism, the gut microbiome, and symptom variability in IBS-M patients.

A few comments: 

77: basal hydrogen concentration in exhaled air were higher in diarrhea period of IBS-M.

"were" should be "was" to agree with the singular subject "concentration": "basal hydrogen concentration in exhaled air was higher in the diarrhea period of IBS-M."

185: Patients were diagnosed according to Rome IV Criteria: The word "was" should be corrected to "were" to maintain subject-verb agreement.

190: The severity of abdominal symptoms were assessed and scored between from 1 to 7 points: The phrase "were assessed and scored between from" is awkwardly worded. You could rephrase it as "The severity of abdominal symptoms was assessed and scored on a scale of 1 to 7 points."

190: what the patients wrote in the diary every day for three month during both constipation and diarrhea period: "What the patients wrote" should be followed by a comma, and "three month" should be corrected to "three months" for grammatical accuracy.

191: Previously, patients were provided with appropriate instructions, and dietary recommendations: It would be clearer to specify who provided the instructions and recommendations. For example, "Patients were provided with appropriate instructions and dietary recommendations by healthcare professionals."

193: Diagnostic test were performed in the fifth week of constipation and then in 5th week of the diarrhea period: "Test" should be pluralized to "Tests", and "5th" should be corrected to "fifth" for consistency.

207: At the beginning, a fasting level of hydrogen in breathing air was measured.

This sentence is clear, but it could be slightly improved by specifying the method used to measure the hydrogen levels (e.g., "At the beginning, the fasting level of hydrogen in the breath was measured using the Gastrolyzer...")

Then, 10 grams lactulose dissolved in 200 mL water was administrated to the patients and breath samples were collected immediately and at 15-minute minute intervals for 3 hours.

209: There is a repetition of "minute" in "15-minute minute intervals". Remove the repeated word for clarity.

The criterion for SIBO positive diagnosis was an increase minimum of 20 ppm of hydrogen within the first 90 minutes of testing.

234: "wave length" should be corrected to "wavelength" for proper spelling.

248: Patients record the type and quantity of products consumed every day for 30 days to investigations in the nutritional diary.

This sentence seems unclear and could be revised for clarity. Perhaps: "Patients recorded the type and quantity of products consumed every day for 30 days in a nutritional diary for investigation purposes."

249: The average daily TRP intake was calculated using the nutritional calculator with application Kcalmar.pro-Premium (Hermex, Lublin, Poland).

"with application" should be revised to "using the application" for smoother readability.

271: In my opinion, there is no such thing as a "Wilcoxon ANOVA test." The Wilcoxon test (Wilcoxon signed-rank test) in my mind is used to compare two paired groups, but ANOVA is used to compare three or more groups. Please explain …

275: I don’t understand … "alternation" may not be the most suitable word in this context. It seems like you might have intended to use "alteration" instead, which means a change or modification ??? Please explain.

Comments on the Quality of English Language

Please improve according to the comments. 

Author Response

Comment (C):

                In my opinion, there is no such thing as a "Wilcoxon ANOVA test." The Wilcoxon test don’t understand … "alternation" may not be the most suitable word in this context. It seems like you might have intended to use "alteration" instead, which means a change or modification ??? Please explain.

Response:

                Thank you very much for pointing out our error. Of course, we agree with your opinion. We used the Wilcoxon test instead of the Student’s t-test because our data did not have a normal distribution. The Wilcoxon test should not be called an ANOVA test.

Comment: I don’t understand … "alternation" may not be the most suitable word in this context. It seems like you might have intended to use "alteration" instead, which means a change or modification ??? Please explain.

Response: ,,…symptoms alternation’’-  it is correct.

Response:  All errors contained in lines: 77, 185, 190, 101, 207, 209, 234, 249  have been corrected  exactly as  recommended.

The order of affiliation has been changed according to the publisher's recommendation.

Thank you very much for jour kindness and linguistic correction of the manuscript !

Round 2

Reviewer 1 Report

Comments and Suggestions for Authors

Disruptions in the redox and tryptophan–kynurenine balance, and alterations in levels of tryptophan and tryptophan metabolites can ‘explain’ the abdominal and systemic symptoms of IBS.

Increased consumption of tryptophan via the kynurenine pathway may result in too little tryptophan and serotonin in both the central nervous system and the gastrointestinal tract.

Most of the serotonin in the body (95%) is found to be in the gastrointestinal tract. Previous  studies have demonstrated that patients with IBS have low serotonin concentrations 10.1016/j.jpsychores.2013.01.008 and small numbers of serotonin-containing neuroendocrine cells in the small intestine.

Impaired small intestinal motor function, as observed in patients with IBS  may thus be a consequence of too low serotonin levels within the intestinal wall. The end product of serotonin catabolism, 5-hydroxy indole acetic acid, is excreted in urine. Previous studies conducted on patients with IBS revealed that it’s excretion is low, indicating low production of serotonin as well. Serotonin is inactivated by reuptake to enterocytes, neurons and platelets, and this serotonin transport function is reduced in patients with IBS.

In my opinion,  the results of this study are not relevant, as the number of patients enrolled is too small, and the criteria of diagnosis are not sufficient,  statistical analysis is not significant, and previous studies published showed different results.

I consider the study should be improved and the results should be reassessed as well.

Comments on the Quality of English Language

Minor English editing required

Author Response

Our team has been conducting research on changes of TRP metabolism since 2006. Especially in functional and inflammatory digestive diseases. The results were published in several journal. These also apply to IBS (10.3390/ijms232315314;  10.3390/nu15081837;  10.3390/nu15051262;  10.3390/ijms25010273), as well as mood disorders ( 10.3390/ijms15040847;  10.3390/nu14153217).   The current study did not assess the patients’ mental status.

The publisher’s regulations limit the posting of our own articles. For this reason, important, but generally known knowledge was limited in this manuscript. The obtained results are consistent with the previous ones and confirm that TRP metabolism is different in different types of IBS. This explains the large discrepancy in the results obtained by other authors.

The value of the results depends not only on the number of respondents , but also on the exact qualification for testing. This is difficult for IBS patients because requires their content to burdensome testing procedures to exclude other diseases.

More recent our studies have shown that limiting or increasing intake of TRP, you can partially change its metabolism and the severity of abdominal complaints. The current results indicate that constant TRP intake does not prevent these changes, and the recurrence  of symptoms of symptoms is not explained.  For this reason, we are currently conducting research on the variability of the gut microbiome, using the FloraGEN GA- map method; research results will be published in 2025.

In response to other comments:

  • The criteria for diagnosis of IBS-M were supplemented.
  • The result of other own research were included in the text.
  • The number of cited articles was increased.
  • Some grammatical errors have been corrected.
  • The statistical evaluation was retained because the methods used were commonly used by other authors, which allows comparison of the results. However, we would be grateful if you could point out other comparable methods useful in this type of research.

Round 3

Reviewer 1 Report

Comments and Suggestions for Authors

All queries have been addressed.

Comments on the Quality of English Language

Minor English editing required.